# Dietary Artemisia Ordosica Polysaccharide Enhances Spleen and Intestinal Immune Response of Broiler Chickens

**DOI:** 10.3390/biology12111390

**Published:** 2023-10-31

**Authors:** Haidong Du, Yuanyuan Xing, Yuanqing Xu, Xiao Jin, Sumei Yan, Binlin Shi

**Affiliations:** College of Animal Science, Inner Mongolia Agricultural University, Hohhot 010018, China; duhaidong1110@163.com (H.D.); xingyuanyuan2014@163.com (Y.X.); happyxyq@yeah.net (Y.X.); yaojinxiao@aliyun.com (X.J.); yansmimau@163.com (S.Y.)

**Keywords:** *Artemisia ordosica*, polysaccharide, feed additive, broiler, immunomodulation

## Abstract

**Simple Summary:**

With the increasing demand for safe livestock and poultry products, antibiotic growth promoters have been banned in several countries, and new alternatives are sought to improve the health of poultry and animals. Plant polysaccharides have been used in functional feed additives as an alternative to antibiotic growth promoters. As a natural bioactive component, *Artemisia ordosica* polysaccharide (AOP) has good activity in immune regulation, as well as antioxidant and anti-inflammatory properties. However, no previous studies have evaluated the response of the immune organs to a diet supplemented with AOP in broiler birds. We hypothesized that AOP supplementation would improve the immune function of broiler chickens. Hence, this study investigated the effects of AOP on the spleen and small intestine immune molecule contents and gene expression in broiler chickens. The present study has revealed that the inclusion of 750 mg/kg AOP led to a significant improvement in immune function and growth performance. This finding suggests that AOP can be used to improve the growth and health of broiler chickens.

**Abstract:**

The spleen and small intestines are the primary immune organs that provide important immunity against various diseases. *Artemisia ordosica* polysaccharide (AOP) could be used as an immunologic enhancer to boost immunity in response to infection. This study was performed to explore the effects of the dietary supplementation of AOP on the growth performance and spleen and small intestine immune function in broilers. A total of 288 AA broilers (1 day old) were randomly assigned into six dietary groups. Each group included six replicates of eight broilers per cage. The broilers were fed with a basal diet supplemented with 0 mg/kg (CON), 50 mg/kg chlortetracycline (CTC), 250, 500, 750, and 1000 mg/kg AOP for 42 d. The results showed that dietary AOP supplementation affected broiler growth performance, with 750 and 1000 mg/kg of AOP being able to significantly improve broiler BWG, and 750 mg/kg of AOP was able to significantly reduce the FCR. The dietary AOP supplementation increased the levels of IgA, IgG, IgM, IL-1β, IL-2, and IL-4 in the spleen and small intestine in a dose-dependent manner (*p* < 0.05). Meanwhile, we found that AOP can promote the mRNA expression of TLR4/MAPK/NF-κB signaling-pathway-related factors (*TLR4*, *MyD88*, *P38 MAPK*, *JNK*, *NF-κB p50*, and *IL-1β*). In addition, the dietary supplementation of 750 mg/kg AOP provides better immunity in the tissue than the CON group but showed no significant difference from the CTC group. Therefore, AOP has an immunoregulatory action and can modulate the immune function of broilers via the TLR4/ NF-ΚB/MAPK signal pathway. In conclusion, dietary supplementation with 750 mg/kg AOP may be alternatives to antibiotics for enhancing broilers’ health, immunity, and growth performance.

## 1. Introduction

Broiler chicken farming is one of the most dynamic sectors of poultry farming due to its high growth rate, short rearing period, low feed costs, and good meat quality [1]. The slaughter age of broilers varies between 4 and 7 weeks [2,3]. Their immune and digestive systems are not completely developed during this period, and their disease susceptibility is high [4]. The high rearing densities used in the modern intensive poultry industry may increase the risk of the broilers’ exposure to various environmental stressors and pathogenic organisms, resulting in reduced growth potential and increased infectivity or fatality case rates, which ultimately lower production yields and cause higher economic losses [5,6].

Antibiotics used to be added to poultry diets for disease control and to improve growth performance [7]. However, the frequent use of antibiotics has increased bacterial resistance and the amount of antibiotic residue in products. As a result, the long-term use of these antibiotics gravely threatens the development of poultry farming and the safety of ecosystems. Therefore, antibiotic growth promoters in the poultry industry have been banned in many countries [8]. At the same time, plant extracts have gained significant attention in enhancing poultry immunity against disease infections. This is mainly attributed to their environmental friendliness, safety, and high efficacy [9].

Plant polysaccharides are biologically active compounds with various biological functions [10]. In recent years, plant polysaccharides have been extensively studied as immune stimulants in both cell and animal models to explore their immune regulatory activities [10,11]. Numerous studies found that the immunomodulatory mechanism of polysaccharides is mainly realized through interactions with the polysaccharide immune cell membrane receptors that trigger various molecular signaling cascades mediated by the cell signaling pathways that promote the secretion of immune factors [12,13]. Immune factors are essential for an animal’s growth performance and health. Therefore, it is imperative to investigate the development of polysaccharides with immunomodulatory properties and their mechanisms of action.

*Artemisia ordosica*, belonging to the Artemisia genus of compositae, is the main perennial semi-shrub plant in the arid and semi-arid areas of northern China. As a medicinal plant, various pharmacological effects of *Artemisia ordosica* have been recorded in traditional Chinese medicine and Mongolian books [14]. *Artemisia ordosica* can be used as an immune enhancer to eliminate dampness and for detoxification and anti-inflammation [14,15]. Recent studies in our laboratory have found that *Artemisia ordosica* extracts have antioxidative, anti-inflammatory, antimicrobial, and growth-promoting effects [15,16,17]. *Artemisia ordosica* polysaccharide (AOP) is an important bioactive component extracted from *Artemisia ordosica*. It has been reported that AOP is regularly employed as a feed supplement to enhance weight gain and promote health by boosting immune regulation and antioxidant capacity in broiler chickens [13]. The spleen is a secondary lymphoid organ composed of immune cells. It is the center of cellular and humoral immunity, crucial to establishing the avian immune system [18,19,20,21]. The intestinal mucosa is an essential mucosal immune system in poultry and is considered the first barrier for resisting the invasion of harmful pathogens [22]. As far as we know, however, there is little research exploring the effects of AOP on immune indices in the spleen and small intestine of broiler chickens. Therefore, based on our previous studies, we have hypothesized that AOP dietary addition can enhance immunity in broilers. To verify this hypothesis, this study was designed to evaluate the effects of the AOP on the immune response in broiler chickens.

## 2. Materials and Methods

This research was conducted at a broiler experimental unit, Inner Mongolia Agricultural University, and all of the experimental procedures were performed in accordance with the national standard Guideline for Ethical Review of Animal Welfare (GB/T 35892–2018) (https://www.chinesestandard.net/PDF.aspx/GBT35892-2018 accessed on 1 August 2023).

### 2.1. Preparation of AOP

*Artemisia ordosica* was harvested from the Ordos district, Inner Mongolia, in July. The water-decoction–ethanol-precipitation method was used to obtain crude *Artemisia ordosica* polysaccharides. Briefly, the grated dry *Artemisia ordosica* was degreased with petroleum ether for 12 h and extracted with water (the ratio of *Artemisia ordosica* powder to water was 1:15) at 60 °C for 4.3 h. The aqueous extract underwent filtration and concentration to reduce its volume to one-fifth of the original. This was followed by precipitation through the addition of 95% ethanol for 48 h, and subsequent centrifugation at 12,000× *g* for 15 min to obtain crude polysaccharides.

The crude polysaccharide underwent sequential washing with petroleum ether, acetone, and ethanol. Subsequently, the crude polysaccharide solution underwent double deproteination using the Sevage method, followed by dialysis against distilled water at 4 °C for 2 days. The detailed extraction steps of the AOP have been previously published [23]. The resulting solution was dried in a vacuum drying chamber. Finally, the purified AOP powder was obtained. Based on the analysis using high-performance liquid chromatography, the monosaccharide composition and molar percentage of the AOP were as follows: arabinose: 6.87%, galactose: 10.67%, glucose: 54.13%, xylose: 2.49%, mannose: 18.37%, galacturonic acid: 4.83% and glucuronic acid: 2.64%.

### 2.2. Experimental Design and Management

A total of 288 AA broilers (1 day old) were randomly allocated to 6 dietary treatments (6 pens/treatment and 8 broilers/pen). The birds were housed in a temperature-controlled room with a window and accommodated in single-level cages consisting of plastic mesh flooring, with eight birds allotted per cage measuring 100 × 50 × 50 cm throughout the duration of the experiment.

A total of 288 AA broilers (1 day old) were randomly allocated 6 dietary treatments (6 pens/treatment and 8 broilers/pen): control group, CON (basal diet); antibiotic group, CTC (basal diet + 50 mg/kg chlortetracycline); *Artemisia ordosica* polysaccharide groups: AOP250 group, AOP500 group, AOP750 group, AOP1000 group, AOP (basal diet + 250, 500, 750, 1000 mg/kg polysaccharide of *Artemisia ordosica*, respectively). The temperature was kept at 32 °C to 34 °C for the first 7 d and then gradually reduced by 3 °C per week until it reached 21 °C. The relative humidity was maintained between 50% and 60%. There were two different types of diets: grower diets (d 1–21) and finisher diets (d 22–42). The maize-soybean-based basal diet (Table 1) was formulated according to the nutritional requirements recommended by the NRC (1994). The BW and feed consumption of the birds per cage were determined on d 0, 21, and 42 to calculate the body weight gain (BWG), feed intake (FI), and feed conversion ratio (FCR).

### 2.3. Sample Collection and Preparation

One chicken per replicate pen, with a BW close to the average BW of each pen, was euthanized by cervical dislocation at 21 and 42 days of age. A total of 36 broilers were used for sample collection. After euthanasia, the middle site of the duodenum, jejunum, ileum (approximately 3 cm), and spleen (approximately 1 g) were immediately taken and rinsed with sterile PBS. The tissues were rapidly frozen in liquid nitrogen and stored at −80 °C until analysis.

### 2.4. Determination of Immune Indexes in Tissue Samples

The frozen spleen, duodenum, jejunum, and ileum samples were weighed at about 0.5 g each, and then homogenized in saline (*w*/*v*, 1:9). The homogenate was centrifuged at 12,000× *g* for 10 min. The resulting supernatant was used to determine the immunoglobulin A (IgA), immunoglobulin G (IgG), immunoglobulin M (IgM), interleukin-1β (IL-1β), interleukin-2 (IL-2), and interleukin-4 (IL-4) concentrations with ELISA kits (Nanjing Jiancheng Bioengineering Institute, Nanjing, China). All procedures were carried out according to the instructions provided by the manufacturer.

### 2.5. Total RNA Extraction and Reverse Transcription

The total RNA from the spleen, duodenum, jejunum, and ileum samples was obtained using TRIzol reagent (TaKaRa, Dalian, China). The RNA was quantified using a Pultton P200CM ultraviolet spectrophotometer (Pultton Technology, San Jose, CA, USA). The total RNA was reverse-transcribed to cDNA using a Prime Script RT reagent kit (TaKaRa, Dalian, China).

### 2.6. Quantitative Real-Time PCR

The real-time PCR was performed with a LightCycler^®^ 96 Real-Time PCR System (Roche, Germany) using the SYBR Premix Ex Taq TM II kit (TAKARA, Dalian, China). The RT-PCR primer information is listed in Table 2. The reactions were as follows: heated at 95 °C for 5 min and a two-step cycle (5 s at 95 °C, 33 s at 60 °C) was repeated for 35 cycles. The mRNA levels were quantified and normalized to β-actin using the 2^−ΔΔCT^ method.

### 2.7. Statistical Analysis

Data were analyzed using one-way analysis of variance (ANOVA) followed by the Tukey–Kramer multiple comparisons test. The linear and quadratic effects of the AOP dose were determined using regression analysis. All data were expressed as mean ± standard error of the mean (SEM). *p* < 0.05 was considered statistically significant.

## 3. Results

### 3.1. Effects of AOP Treatment on the Growth Performance in Broilers

The growth performance results of broilers were shown in Table 3. During d 22 to 42, the BWG of the broilers was higher in the AOP750-supplemented group in comparison to those fed CON or AOP250-supplemented diets (*p* < 0.05), and the BWG linearly (*p* = 0.01) or quadratically (*p* < 0.05) increased with the supplemental AOP level.

During the overall period (d 0 to 42), the BWG of the broilers fed 750 or 1000 mg/kg AOP-supplemented diets was higher in comparison to those fed CON or 250 mg/kg AOP-supplemented diets (*p* < 0.05), and the BWG linearly (*p* = 0.01) or quadratically (*p* < 0.05) increased with the supplemental AOP level. The AOP750 group had a lower FCR than the CON, AOP250 and AOP500 groups (*p* < 0.05), but there were no differences in FCR between the AOP750 group and CTC group (*p* > 0.05). The FCR linearly (*p* < 0.01) or quadratically (*p* < 0.05) decreased with the supplemental AOP level. There was no significant difference in the FI among the groups (*p* > 0.05).

### 3.2. Effects of AOP Treatment on the Spleen Immunoglobulins in Broilers

The results of the immunoglobulin in the spleen are shown in Table 4. On d 21, the supplementation with AOP at 750 mg/kg increased the concentration of IgG compared to the 1000 mg/kg AOP, CON and CTC groups (*p* < 0.05), and the IgG content quadratically increased with the supplemental AOP level on d 21 (*p* < 0.05). There were no significant differences in the IgA and IgM concentration among the treatments on d 21 (*p* > 0.05).

On d 42, dietary inclusion of 750 mg/kg AOP resulted in a higher concentration of IgA when compared to the CON and CTC groups (*p* = 0.01), and the IgA content quadratic increased with the increase in the AOP additive amount (*p* = 0.05). Dietary inclusion of 250 and 750 mg/kg AOP resulted in a higher concentration of IgG compared to the CON group (*p* < 0.05). Dietary inclusion of 750 and 1000 mg/kg AOP resulted in a higher concentration of IgM compared to the CON group (*p* < 0.05), but there were no differences in the IgM activities among the AOP750, AOP1000 and CTC groups. The IgG and IgM contents of the spleen showed a linear or quadratic increase with the increasing inclusion level of AOP in the diet (*p* < 0.05).

### 3.3. Effects of AOP Treatment on the Small Intestine Immunoglobulins in Broilers

Table 5 shows the concentration of immunoglobulins in the small intestine. On d 21, compared to the CON group, the AOP1000 group significantly increased the concentration of IgA in the ileum (*p* < 0.05), and the concentration of IgG in the jejunum (*p* < 0.01), and there was no significant difference between the CTC group. The CON and AOP groups could increase jejunum IgM activity compared to the CTC group (*p* < 0.01). The activities of IgG and IgM in the duodenum and ileum, however, showed no significant difference among the groups (*p* > 0.05). With increasing AOP levels, the production of ileum IgA increased in a linear or quadratic manner (*p* < 0.01). The IgA content of the duodenum showed a linear or quadratic decrease with the inclusion level of AOP in the diet (*p* < 0.05), while the IgG content of the jejunum decreased in a quadratic manner (*p* = 0.01).

On d 42, the duodenum IgA content was significantly higher in the AOP500 and AOP750 groups than in the CON group (*p* = 0.05), and the jejunum IgA content in AOP750 group was higher than the CTC or CON group (*p* = 0.05). Supplementation with 500 mg/kg AOP increased the ileum IgA content (*p* = 0.01) and duodenum IgG content (*p* < 0.01) more than the CON and CTC groups. The AOP500 group exhibited markedly elevated IgG contents in the jejunum compared to the AOP250, CTC, and CON groups (*p* < 0.05). Supplementation with 500 mg/kg AOP increased the ileum IgG content more than the CTC, AOP750 and AOP1000 group (*p* < 0.05). The AOP250, AOP750, and AOP1000 groups exhibited markedly elevated duodenum IgM contents compared to the CON group (*p* < 0.05); however, there was no significant difference between the AOP and CTC groups. Dietary inclusion of 250–1000 mg/kg AOP resulted in a higher concentration of jejunum IgM (*p* = 0.01), and dietary inclusion of 250–500 mg/kg AOP resulted in a higher concentration of ileum IgM (*p* = 0.01) when compared to the CTC group. The results of the regression analysis showed that the contents of the jejunum IgG, ileum IgG and IgM and duodenum IgA increased linearly and quadratically with the increase in AOP (*p* < 0.05), while the IgG content of the duodenum and the IgA content of the ileum increased in a quadratic manner (*p* < 0.01), and the IgM content of the duodenum increased in a linear manner (*p* < 0.05).

### 3.4. Effects of AOP Treatment on the Spleen Cytokines in Broilers

Table 6 showed the concentration of cytokines in the spleen. On d 21, supplementation with 750–1000 mg/kg AOP increased the IL-1β content more than the CON, CTC, AOP250 and AOP500 group (*p* = 0.01) and the levels of IL-1β increased linearly and quadratically with an increasing AOP dose (*p* < 0.01). Activities of IL-2 and IL-4 in the spleen, however, showed no difference among the groups.

On d 42, dietary inclusion of 1000 mg/kg AOP resulted in a higher IL-1β and IL-2 content when compared to the CON group (*p* < 0.05), while dietary inclusion of 750 and 1000 mg/kg AOP resulted in a higher IL-4 content when compared to the CON group (*p* < 0.05). With increasing AOP levels, the production of IL-1β, IL-2, and IL-4 increased in a linear or quadratic manner (*p* < 0.05).

### 3.5. Effects of AOP Treatment on the Small Intestine Cytokines in Broilers

The results of small intestine cytokines are shown in Table 7. On d 21, supplementation with 500 and 1000 mg/kg AOP increased duodenum IL-1β activity more than the CON group (*p* < 0.05), and its duodenum IL-1β activity was similar to that in the CTC group. The AOP250, AOP500 and CON group had a higher jejunum IL-1β content than the CTC group (*p* < 0.01). The AOP500, AOP750 and AOP1000 group had a higher duodenum IL-2 content than the CON group (*p* < 0.01). The AOP750 and AOP1000 groups had a higher ileum IL-2 content than the CON group (*p* < 0.01). Supplementation with 1000 mg/kg AOP increased jejunum IL-2 activity more than the CTC group (*p* < 0.01). Supplementation with 250, 500, and 1000 mg/kg AOP increased duodenum IL-4 activity more than the CON group (*p* < 0.05), and its duodenum IL-4 activity was similar to that in the CTC group. In addition, the results of the regression analysis showed that the levels of IL-1β in the duodenum and IL-1β and IL-2 in the ileum increased linearly and quadratically with the increase in the AOP additive (*p* < 0.05), while the levels of IL-1β in the jejunum and IL-2 in the duodenum increased in a linear manner (*p* < 0.05), and the levels of IL-2 in the jejunum decreased in a quadratic manner (*p* < 0.05).

On d 42, the AOP500 group showed a higher level of ileum IL-1β than the other group (*p* < 0.01). The AOP1000 group showed a higher level of jejunum IL-1β than the CTC group (*p* < 0.01). Dietary inclusion of 500 mg/kg AOP resulted in a higher concentration of duodenum IL-1β compared to the CTC group (*p* < 0.01), but there were no differences in duodenum IL-1β activities among the AOP500 group and CON group. The AOP500 group showed a higher level of ileum IL-2 than the other groups (*p* = 0.01). The AOP250 group showed a higher level of duodenum IL-4 than the CTC group (*p* < 0.01). The activities of jejunum and duodenum IL-2, jejunum and ileum IL-4, however, showed no difference among the groups. The IL-1β and IL-4 contents of the duodenum showed a linear or quadratic decrease with the inclusion level of AOP in the diet (*p* < 0.01). The IL-1β and IL-4 contents of the ileum increased linearly or quadratically with the addition of AOP in the diet (*p* < 0.05).

### 3.6. Effects of AOP Treatment on the Spleen TLR4/NF-κB/MAPK Pathway-Related Gene Expressions in Broilers

Table 8 showed the mRNA expressions of the genes related to *TLR/NF-κB/MAPK* signaling pathway in the spleen. On d 21, the AOP1000 group effectively elevated the mRNA expression *of TLR4*, *MyD88*, *NF-κB p50* and *IL-1β* compared to the CON group (*p* < 0.05). The AOP750 group showed a higher level of *JNK* expression than the CTC group (*p* < 0.01). The *P38 MAPK* expression in the spleen, however, showed no significant difference among the groups (*p* > 0.05). The mRNA levels of *TLR4*, *MyD88* and *IL-1β* increased linearly or quadratically with an increasing AOP dose (*p* < 0.05), and the *NF-κB p50* mRNA level increased linearly with an increasing AOP dose (*p* < 0.05).

On d 42, the AOP250, AOP500 and AOP1000 group showed a higher level of *TLR4* expression than the CON group (*p* < 0.01). The mRNA expression level of *MyD88*, *NF-κB p50*, and *IL-1β* was elevated in the AOP1000 group compared with the CON group (*p* < 0.05). The mRNA expression level of *P38 MAPK* was more elevated in the AOP groups than in the CON group (*p* < 0.01). The AOP750 group showed higher *JNK* mRNA expression levels than the CON group (*p* = 0.01), but there were no differences in the *JNK* mRNA expression level between the AOP750 group and CTC group. The results of the regression analysis showed that the mRNA levels of *NF-κB/p50*, *JNK* and *IL-1β* increased linearly or quadratically as the AOP dose increased (*p* < 0.05), while the *TLR4* and *P38 MAPK* mRNA level increased quadratically with an increasing AOP dose (*p* < 0.05).

### 3.7. Effects of AOP Treatment on the Duodenum TLR4/NF-κB /MAPK-Pathway-Related Gene Expressions in Broilers

Table 9 showed relative mRNA expressions of genes in the duodenum. On d 21, the *TLR4* mRNA expression in the AOP250, AOP500, AOP750 groups, the *MyD88* mRNA expression in all AOP groups, and the *NF-κB p50* mRNA expression in the AOP500 group were markedly higher than the CON group (*p* < 0.05). The mRNA expression of *P38 MAPK* and *IL-1β* was markedly higher in the AOP1000 group than in the CON group (*p* < 0.01), but there were no differences in the *P38 MAPK* and *IL-1β* mRNA expression level between the AOP1000 group and CTC group. The CTC group increased the *JNK* mRNA expression level more than the CON group (*p* < 0.05), but there were no differences in the *JNK* mRNA expression level between the AOP750, AOP1000 group and CTC group. The results of the regression analysis showed that the mRNA levels of *TLR4*, *MyD88*, *P38 MAPK*, *JNK*, and *IL-1β* increased linearly or quadratically as the AOP increased (*p* < 0.05).

On d 42, the AOP500, AOP750 and AOP1000 groups exhibited a higher level of *TLR4* mRNA expression than the CON group (*p* < 0.05). Compared with the CON group, the AOP750 andAOP1000 group exhibited a higher level of *MyD88* expression (*p* < 0.05), but there were no differences in the *MyD88* mRNA expression level among the AOP750, AOP1000 group and CTC group. The mRNA expression level of *NF-κB p50* exhibited a significant increase for the CTC groups compared to the CON group (*p* < 0.05). However, no discernible differences were observed in the *NF-κB p50* expression level among the AOP500, AOP750, AOP1000 group and CTC group. The AOP750 group showed a higher level of *JNK* expression than the CON group (*p* < 0.01). The dietary supplementation with 1000 mg/kg AOP increased the *IL-1β* mRNA expression more than in the CON group (*p* = 0.05). The mRNA expression of *P38 MAPK* in duodenum, however, showed no significant difference among the groups (*p* > 0.05). The results of the regression analysis showed that the mRNA levels of *TLR4*, *MyD88*, *NF-κB/p50* and *IL-1β* increased linearly or quadratically as the AOP dose increased (*p* < 0.05), while the *JNK* mRNA level increased linearly with an increasing AOP dose (*p* < 0.05).

### 3.8. Effects of AOP Treatment on the Jejunum TLR4/NF-κB/MAPK-Pathway-Related Gene Expressions in Broilers

Table 10 showed the relative mRNA expressions of genes in the jejunum. On d 21, the AOP500 group showed a higher level of *IL-1β* expression than the CON, CTC, AOP750, and AOP1000 groups (*p* < 0.01), and showed a linear effect with an increasing AOP dose. There was no significant difference in the expression of *TLR4*, *MyD88*, *NF-κB p50*, *JNK*, and *P38 MAPK* in the jejunum among the different treatments (*p* > 0.05).

On d 42, supplementation with 750 mg/kg AOP increased the *TLR4*, *MyD88*, *NF-κB p50*, and *IL-1β* mRNA expression level more than the CON and CTC groups (*p* < 0.05). The *P38 MAPK* mRNA expression was significantly increased in the AOP250 and AOP750 groups compared to the CTC group (*p* < 0.01). The *JNK* mRNA expression was significantly increased in all AOP groups compared to the CON group (*p* < 0.01) but was not significantly different from the CTC group. The regression analysis results showed that the *TLR4*, *MyD88*, *JNK* and *IL-1β* mRNA levels increased linearly or quadratically as the AOP dose increased (*p* < 0.05).

### 3.9. Effects of AOP Treatment on the Ileum TLR4/NF-κB /MAPK-Pathway-Related Gene Expressions in Broilers

Table 11 showed the gene expressions levels in the ileum. On d 21, the mRNA expression of *TLR4* and *IL-1β* in the AOP500 group, the *MyD88* mRNA expression in the AOP500 and AOP750 groups, and *NF-κB p50* mRNA expression in the AOP500, AOP750, AOP1000 groups were significantly higher than the CON group (*p* < 0.05). Supplementation with 250, 500, and 750 mg/kg AOP increased the *JNK* mRNA expression level compared to the CTC groups (*p* < 0.01). The results of the regression analysis showed that the mRNA levels of *NF-κB/p50* increased linearly or quadratically as the AOP dose increased (*p* < 0.01), and the *JNK* mRNA level decreased linearly or quadratically with increasing AOP doses (*p* < 0.01). The *MyD88* mRNA level increased quadratically with an increasing AOP dose (*p* = 0.01).

On d 42, the mRNA expression of *TLR4* in the AOP750 and AOP1000 groups, the MyD88 mRNA expression in the AOP500, AOP750, AOP1000 groups, and the *NF-κB p50* mRNA expression in AOP500 and AOP groups were significantly higher than the CON group (*p* < 0.05) but were not significantly different from those of the CTC group. The AOP500 and AOP750 groups showed a higher level of *IL-1β* expression than the CON and CTC groups (*p* < 0.01). The mRNA expression of *P38 MAPK* and *JNK*, however, showed no significant difference among the groups (*p* > 0.05). The results of the regression analysis showed that the expression levels of *TLR4*, *MyD88*, and *IL-1β* increased linearly or quadratically as the AOP dose increased (*p* < 0.05), and the *NF-κB/p50* mRNA level increased quadratically with an increasing AOP dose (*p* < 0.01).

## 4. Discussions

During d 22 to d 42 and d 0 to 42, the broilers in the AOP750 group significantly increased BWG and decreased FCR, and showed a similar BWG and FCR with those in the CTC group. This finding indicated that the 750 mg/kg AOP dose could be an alternative to antibiotic growth promoters in poultry diets. Similar results were found by Wang et al. when the broiler chicken diet was supplemented with *Astragalus* polysaccharides, which significantly increased the ADG and reduced the feed-to-gain ratio of broilers compared with the control group, and there was no significant difference compared with the antibiotic group [24]. Several studies have reported that the mode of action for enhancing growth performance by the plant polysaccharide might be related to enhancing immune functions, ameliorating gut morphology, and modulating the intestinal microbials of the birds [3,7,8]. In this study, AOP supplementation increased the spleen and small intestinal tissue immune factor levels. This finding suggests that AOP may play a role in regulating the immune system in chickens. Additionally, in our previous publication, we found that AOP can improve the intestinal morphology by increasing the intestinal villi height and decreasing the crypt depth; furthermore, the effects seen in the AOP750 group were better than other groups [25]. In addition, the 750 mg/kg AOP dose was found to significantly reduce the number of harmful bacteria (*Escherichia coli*) and increase the number of beneficial bacteria (*Bifidobacterium*) in the cecal contents of broiler chickens [25]. These findings suggest that AOP may have positive effects on intestinal well-being in broiler chickens. In conclusion, dietary supplementation with 750 mg/kg AOP improves gastrointestinal health by regulating the immune system, improving intestinal morphology and modulating intestinal microbiota, thus improving overall health and growth performance of broiler chickens. Furthermore, the present study found that the broilers’ growth stage may influence the AOP’s effect on growth performance and immune function.

In this study, the effects of the AOP on growth performance were only observed during the d 22 to 42 stage. The AOP treatments did not affect the broilers’ BWG, FI or FCR from d 0 to 21. At the same time, the AOP supplementation had a poor immunomodulatory capacity in the broilers during the experimental period of 0-21d. Similar results were also found by Zhang et al. [26]. This may be due to the broiler experiencing rapid growth at an early stage, as the intestinal microbiota and the immune system still need to develop fully. Consequently, plant polysaccharides have a lower growth-promoting effect in the early stages than in the late stages of broiler growth. In addition, the period of AOP supplementation of 21d could be too short to show effects.

Poultry adaptive immunity is divided into humoral and cellular immunity [27]. Humoral immunity is the primary immunity mediated by immunoglobulin [28]. Immunoglobulin is an antigen-specific receptor that binds with the corresponding antigen, signaling to the immune cells to initiate effective adaptive immune responses [29,30,31]. Previous studies have reported that plant polysaccharides enhance immunity by promoting the secretion of immunoglobulins in the spleen or intestinal epithelial cells [32,33]. A study found that oral Korean *persimmon vinegar* polysaccharides significantly increased IgA levels in the small intestine fluid of mice [34]. Furthermore, experiments conducted in vitro have demonstrated that the application of Korean *persimmon vinegar* polysaccharides to small intestinal Peyer’s patch cells resulted in a dose-dependent increase in IgA levels [34]. Wang et al. (2008) found that *Astragalan*, *Ganoderan*, and *Lentinan* had immunopotentiating effects by promoting the spleen lymphocyte secretion of IgG [35]. IgA, IgG and IgM are three classes of major immunoglobulins in poultry [30]. In this study, the AOP significantly induced immunoglobulin secretion in the spleen and small intestine tissues. Compared to the CON and CTC groups, the content of IgA, IgG, and IgM increased in the 750–1000 mg/kg AOP group. Similar results were found by Zhang et al. [36], which showed that subcutaneously administered *Artemisia rupestris* L. aqueous extracts could increase the titers of antigen-specific antibodies (IgG, IgG1, and IgG2a) in ICR mice. The above results may be due to the plant polysaccharides enhancing the secretion of cytokines, which stimulate the proliferation and differentiation of immune cells and the production of autoantibodies.

Cytokines are key regulators in orchestrating mucosal immunity and play an essential role in inflammatory responses [37]. IL-1β is an important regulator that stimulates immune system development and differentiation and promotes the activation and polarization of T cells [38]. IL-2 promotes the activation of natural killer cells, B lymphocytes, and T lymphocytes, and the production of antibodies, while the main functions of IL-4 are to stimulate the proliferation and differentiation of T cells and B cells, and to produce immunoglobulin [39,40]. Yang et al. (2020) found that *Artemisia rupestris* L. aqueous extracts (of which the major component is polysaccharide) significantly increased the number of CD4^+^ and CD8^+^ T cells, stimulated the production of IL-1β and IL-6, and increased the expression of the costimulatory molecules of dendritic cells, thereby improving the regulatory role of immune cells [41]. Our previous study showed that *Artemisia argyi* polysaccharide at an appropriate dose could increase the contents of immunoglobulin and interleukin in the culture medium of broilers’ peripheral blood lymphocytes [42]. In this study, it was shown that the contents of IL-1β, IL-2, and IL-4 in the spleen and small intestine of broilers were increased by adding different doses of AOP, suggesting that AOP might be able to be used as an immune stimulator to regulate the immune response of broilers. In addition, the contents of cytokines in the tissues of broilers in the 500–1000 mg/kg AOP addition groups were not significantly different from those in the CTC group, nor were they higher than those of the CTC group, suggesting that an appropriate dose of AOP is expected to replace the role of CTC and that it plays a regulatory role in immune function.

To further investigate the immunoregulatory mechanism of AOP in broilers, we investigated the expression of *TLR4* and its downstream *NF-κB* and *MAPK* signaling pathway-related genes. TLR4 has been identified to be an important membrane receptor for mediating the effects of plant polysaccharides on the activation of immune cells [43,44]. Many studies have shown that plant polysaccharides can activate the NF-κB and MAPK signaling pathways through TLR4 receptors, and promote the release of related cytokines, and exert their immunoregulatory functions [45,46,47]. Yang et al. (2020) found that *Artemisia* polysaccharides can activate the MAPK and NF-κB signaling pathways mediated by TLR4/TLR2 and significantly improve the immunomodulatory activity of immune cells [41]. The present study found that the addition of 500–1000 mg/kg of AOP to the broiler diet improved the *TLR4* gene expression levels in the spleen and small intestine tissue to varying degrees. This suggests that TLR4 is capable of recognizing AOP. Additionally, the gene expression levels of *MyD88*, *P38 MAPK*, *JNK*, *NF-κB p50* and *IL-1β*, which lie downstream of *TLR4*, were significantly increased in different tissues, indicating that TLR4 can bind to AOP and transmit extracellular signals to activate the NF-κB and MAPK pathways. Similar results have been found in other studies of plant polysaccharides. Talapphet et al. (2021) found that polysaccharides isolated from *Taraxacum platycarpum* showed immunostimulatory properties by activating TLR4/TLR2/MAPK/NF-κB pathways to upregulate the mRNA expression levels of *IL-1β*, *IL-6*, *TNF-α*, and *IL-10* in RAW264.7 cells [48]. Xie et al. (2020) found that the *Alfalfa* polysaccharide could induce *P38 MAPK*, ERK, and JNK phosphorylation, and the NF-κB p65 nuclear subunit translocation via the splenic B cell membrane receptor TLR4, and then activate the immune functions through the MAPK/NF-κB signaling pathway [45]. Additionally, it has been found that AOP can reduce the LPS-induced overactivation of TLR4 and its downstream inflammatory factors in chicken peripheral blood lymphocytes and liver tissue [13,49]. The above results show AOP exhibits good immunomodulatory capacities under normal physiological conditions, improving the body’s immune function via activating the TLR4/NF-κB/MAPK signal pathway. However, when the body is under acute stress, the antagonistic effect of AOP on pathogens may prevent the over-excitation of the immune pathway.

## 5. Conclusions

In summary, broiler diets supplemented with AOP at the 750 mg/kg level can promote growth performance and regulate immune function. The above results have confirmed our hypothesis. In addition, the present study has demonstrated that broiler diets supplemented with AOP could be applied to improve immune function partly through the TLR4/NF-κB/MAPK signal pathway, promoting the production of immune factors in the tissues of broilers. Thus, the supplementation of AOP as a potential natural immunomodulatory agent, as well as an antibiotic substitute, can be another strategy to improve the health and growth performance of broilers.

## Figures and Tables

**Table 1 biology-12-01390-t001:** Composition and nutrient levels of basal diets (air-dry basis, %).

Ingredients	1–21 Days of Age	22–42 Days of Age
Corn	52.50	58.80
Soybean meal	40.00	33.80
Soybean oil	3.00	3.00
Dicalcium phosphate	1.90	1.80
Limestone	1.08	1.22
Salt	0.37	0.37
Lysine	0.05	0.03
Methionine	0.19	0.07
Premix ^(a)^	0.80	0.80
Choline	0.11	0.11
Total	100.00	100.00
Nutrient levels ^(b)^		
Metabolizable energy (MJ/kg)	12.42	12.62
Crude protein	21.77	19.65
Calcium	1.00	1.02
Available phosphorus	0.44	0.42
Lysine	1.34	1.15
Methionine	0.55	0.40
Cystine	0.40	0.36

^(a)^ Premix provided the following per kilogram of diet: vitamin A 9000 IU; vitamin D_3_ 3000 IU; vitamin E 26 IU; vitamin K_3_ 1.20 mg; vitamin B_1_ 3.00 mg; vitamin B_2_ 8.00 mg; vitamin B_6_ 4.40 mg; vitamin B_12_ 0.012 mg; niacin 45 mg; calcium pantothenate 15 mg; folic acid 0.75 mg; biotin 0.20 mg; choline chloride 1100 mg; Fe 100 mg; Cu 10 mg; Zn 108 mg; Mn 120 mg; I 1.5 mg; Se 0.35 mg. ^(b)^ Crude protein was the measured value, while the others were all calculated values.

**Table 2 biology-12-01390-t002:** Primer sequences and parameter.

Genes	GenBank Accession No	Primer Sequences, 5′-3′	Length, bp
* TLR4 *	NM_001030693	F-TTCAGAACGGACTCTTGAGTGG	131
		R-CAACCGAATAGTGGTGACGTTG	
*MyD88*	NM_001030962	F-CCTGGCTGTGCCTTCGGA	198
		R-TCACCAAGTGCTGGATGCTA	
*P38 MAPK*	AJ719744.1	F-TGTGTTCACCCCTGCCAAGT	149
		R-GCCCCCGAAGAATCTGGTAT	
* JNK *	AB000807.1	F-AGCAGCCTCGATGCCTTGAC	110
		R-CAAGCAATTCAGGCCCAATG	
* NF-κB p50 *	NM_205134	F-GAAGGAATCGTACCGGGAACA	80
		R-CTCAGAGGGCCTTGTGACAGTAA	
* IL-1β *	NM_204524	F-CAGCCTCAGCGAAGAGACCTT	84
		R-ACTGTGGTGTGCTCAGAATCC	
* β-actin *	NM_205518	F-GCCAACAGAGAGAAGATGACAC	118
		R-GTAACACCATCACCAGAGTCCA	

*TLR4*, toll-like receptor 4; *MyD88*, myeloid differentiation factor 88; *P38 MAPK*, P38 mitogen-activated protein kinase; *JNK*, C-Jun N-terminal kinase; *NF-κB p50*, nuclear factor kappa B p50; *IL-1β*, interleukin 1 beta; *β-actin*, beta-actin; F, forward primer; R, reverse primer.

**Table 3 biology-12-01390-t003:** Effects of AOP on growth performance in the spleen of broilers.

Items	AOP Supplemental Level (mg/kg)					CTC (mg/kg)		*p*-Value		
	0	250	500	750	1000	50	SEM	ANOVA	Linear	Quadratic
0–21 d										
BWG, g	569	611	612	612	616	584	23.94	0.67	0.22	0.33
FI, g	867	868	847	868	880	845	23.94	0.89	0.67	0.78
FCR	1.52	1.42	1.39	1.42	1.43	1.45	0.03	0.26	0.18	0.06
22–42 d										
BWG, g	1439 ^b^	1452 ^b^	1466 ^ab^	1563 ^a^	1529 ^ab^	1529 ^ab^	30.45	0.03	0.01	0.02
FI, g	2843	2990	2973	2988	2870	2922	97.65	0.72	0.55	0.58
FCR	2.04	1.99	2.09	2.02	2.01	1.91	0.03	0.06	0.24	0.51
0–42 d										
BWG, g	2096 ^c^	2104 ^c^	2168 ^bc^	2280 ^a^	2250 ^ab^	2228 ^ab^	27.30	0.04	0.01	0.02
FI, g	3562	3691	3654	3755	3656	3668	114.66	0.93	0.55	0.66
FCR	1.83 ^ab^	1.82 ^ab^	1.83 ^ab^	1.69 ^c^	1.73 ^bc^	1.73 ^bc^	0.02	0.04	<0.01	0.03

BWG, body weight gain; FI, feed intake; FCR, feed conversion ratio. ^abc^ Different letters in the same row denote significant differences between treatments.

**Table 4 biology-12-01390-t004:** Effects of AOP on immunoglobulins in the spleen of broilers.

Items	AOP Supplemental Level (mg/kg)					CTC (mg/kg)		*p*-Value		
	0	250	500	750	1000	50	SEM	ANOVA	Linear	Quadratic
21 d										
IgA, μg/mg prot.	7.89	6.57	7.84	8.35	8.16	7.47	0.88	0.76	0.37	0.62
IgG, μg/mg prot.	47.88 ^b^	50.75 ^ab^	58.81 ^ab^	68.42 ^a^	51.53 ^b^	45.76 ^b^	4.19	0.02	0.35	0.04
IgM, μg/mg prot.	16.51	16.68	15.97	15.89	14.94	13.46	1.39	0.61	0.31	0.57
42 d										
IgA, μg/mg prot.	6.12 ^b^	6.75 ^b^	6.78 ^b^	8.23 ^a^	5.90 ^b^	6.36 ^b^	0.32	0.01	0.96	0.05
IgG, μg/mg prot.	49.01 ^b^	57.89 ^a^	54.14 ^ab^	60.69 ^a^	56.20 ^ab^	55.45 ^ab^	2.24	0.04	0.05	0.03
IgM, μg/mg prot.	14.23 ^b^	15.39 ^ab^	15.23 ^ab^	18.01 ^a^	17.15 ^a^	17.43 ^a^	0.90	0.04	0.01	0.04

IgA, immunoglobulin A; IgG, immunoglobulin G; IgM, immunoglobulin M. ^abc^ Different letters in the same row denote significant differences between treatments.

**Table 5 biology-12-01390-t005:** Effects of AOP on immunoglobulins in the small intestine of broilers.

Items		AOP Supplemental Level (mg/kg)					CTC (mg/kg)		*p*-Value		
		0	250	500	750	1000	50	SEM	ANOVA	Linear	Quadratic
21 d											
IgA	Duodenum	5.93	6.10	5.48	5.22	4.98	5.42	0.4	0.37	0.02	0.05
µg/mg prot.	Jejunum	3.62	4.05	3.69	3.19	3.76	4.20	0.45	0.74	0.89	0.98
	Ileum	7.70 ^b^	7.41 ^b^	8.12 ^ab^	9.36 ^ab^	10.24 ^a^	8.57 ^ab^	0.57	0.05	<0.01	<0.01
IgG	Duodenum	43.86	42.15	43.93	41.10	35.18	36.64	2.93	0.25	0.06	0.08
µg/mg prot.	Jejunum	39.56 ^bc^	36.95 ^bc^	30.00 ^c^	31.31 ^c^	53.44 ^a^	43.76 ^ab^	3.13	<0.01	0.15	0.01
	Ileum	66.61	56.10	70.40	70.72	65.12	56.94	5.31	0.50	0.65	0.86
IgM	Duodenum	17.58	17.00	16.70	15.59	15.13	15.12	1.1	0.98	0.48	0.17
µg/mg prot.	Jejunum	18.43 ^a^	18.99 ^a^	15.32 ^a^	15.33 ^a^	14.79 ^a^	9.77 ^b^	1.39	<0.01	0.06	0.14
	Ileum	28.57	20.80	34.60	26.45	32.95	25.85	4.49	0.45	0.38	0.67
42 d											
IgA	Duodenum	5.60 ^b^	5.73 ^b^	8.81 ^a^	8.73 ^a^	7.59 ^ab^	6.80 ^ab^	0.82	0.05	0.04	0.04
µg/mg prot.	Jejunum	5.37 ^bc^	5.81 ^abc^	6.44 ^ab^	7.17 ^a^	5.61 ^abc^	4.64 ^c^	0.45	0.05	0.44	0.15
	Ileum	7.01 ^b^	8.94 ^a^	9.10 ^a^	6.94 ^b^	6.13 ^b^	7.02 ^b^	0.49	0.01	0.11	<0.01
IgG	Duodenum	65.73 ^b^	73.18 ^ab^	81.60 ^a^	73.12 ^ab^	52.96 ^c^	50.00 ^c^	3.38	<0.01	0.10	<0.01
µg/mg prot.	Jejunum	42.99 ^b^	44.43 ^b^	53.51 ^a^	49.77 ^ab^	51.83 ^ab^	43.44 ^b^	2.55	0.05	0.02	0.05
	Ileum	57.18 ^a^	52.71 ^ab^	65.96 ^a^	40.32 ^b^	41.68 ^b^	39.01 ^b^	4.08	<0.01	0.02	0.05
IgM	Duodenum	15.00 ^b^	20.18 ^a^	19.15 ^ab^	19.68 ^a^	20.82 ^a^	16.69 ^ab^	1.10	0.05	0.04	0.06
µg/mg prot.	Jejunum	16.25 ^a^	15.51 ^a^	18.55 ^a^	15.40 ^a^	15.27 ^a^	11.27 ^b^	1.09	0.01	0.70	0.57
	Ileum	23.86 ^ab^	28.27 ^a^	27.13 ^a^	19.01 ^b^	17.55 ^b^	17.89 ^b^	2.08	0.01	0.05	0.02

IgA, immunoglobulin A; IgG, immunoglobulin G; IgM, immunoglobulin M. ^abc^ Different letters in the same row denote significant differences between treatments.

**Table 6 biology-12-01390-t006:** Effects of AOP on cytokines in the spleen of broilers.

Items	AOP Supplemental Level (mg/kg)					CTC (mg/kg)		*p*-Value		
	0	250	500	750	1000	50	SEM	ANOVA	Linear	Quadratic
21 d										
IL-1β, pg/mg prot.	8.06 ^b^	7.83 ^b^	8.28 ^b^	11.2 ^a^	12.81 ^a^	7.84 ^b^	0.69	0.01	<0.01	<0.01
IL-2, pg/mg prot.	7.95	7.71	7.65	7.21	7.5	6.83	0.69	0.5	0.62	0.43
IL-4, pg/mg prot.	2.19	2.11	2.16	2.15	1.95	1.72	0.16	0.45	0.42	0.66
42 d										
IL-1β, pg/mg prot.	7.84 ^c^	9.03 ^bc^	9.49 ^bc^	11.37 ^ab^	14.37 ^a^	12.60 ^a^	0.98	<0.01	<0.01	<0.01
IL-2, pg/mg prot.	7.11 ^b^	7.58 ^ab^	7.67 ^ab^	7.79 ^ab^	8.21 ^a^	8.11 ^a^	0.53	0.04	<0.01	<0.01
IL-4, pg/mg prot.	1.68 ^b^	1.82 ^ab^	1.79 ^ab^	2.14 ^a^	2.17 ^a^	2.15 ^a^	0.09	0.04	0.04	0.05

IL-1β, interleukin 1 beta; IL-2, interleukin 2; IL-4, interleukin 4. ^abc^ Different letters in the same row denote significant differences between treatments.

**Table 7 biology-12-01390-t007:** Effects of AOP on cytokines in the small intestine of broilers.

Items		AOP Supplemental Level (mg/kg)					CTC (mg/kg)		*p*-Value		
		0	250	500	750	1000	50	SEM	ANOVA	Linear	Quadratic
21 d											
IL-1β	Duodenum	3.98 ^b^	4.24 ^ab^	5.40 ^a^	4.65 ^ab^	5.60 ^a^	5.34 ^a^	0.35	0.03	0.01	0.03
pg/mg prot.	Jejunum	8.92 ^a^	9.25 ^a^	8.46 ^a^	6.74 ^b^	7.81 ^ab^	6.53 ^b^	0.41	<0.01	0.02	0.08
	Ileum	9.56	9.99	11.34	11.37	12.39	9.56	0.41	0.20	0.02	0.05
IL-2	Duodenum	4.45 ^c^	5.03 ^bc^	5.96 ^a^	5.67 ^a^	5.96 ^a^	5.44 ^ab^	0.25	<0.01	0.03	0.06
pg/mg prot.	Jejunum	5.24 ^ab^	4.96 ^ab^	4.72 ^ab^	3.76 ^c^	5.41 ^a^	4.31 ^bc^	0.24	<0.01	0.44	0.03
	Ileum	6.53 ^d^	6.78 ^cd^	7.45 ^bcd^	8.77 ^a^	8.31 ^ab^	7.74 ^abc^	0.27	<0.01	<0.01	<0.01
IL-4	Duodenum	1.69 ^b^	2.07 ^a^	2.12 ^a^	1.76 ^ab^	2.05 ^a^	1.87 ^ab^	0.01	0.04	0.13	0.15
pg/mg prot.	Jejunum	1.93	1.52	1.50	1.69	1.85	1.48	0.14	0.15	0.99	0.07
	Ileum	2.70	2.43	2.72	2.59	2.84	2.56	0.20	0.87	0.59	0.67
42 d											
IL-1β	Duodenum	7.24 ^a^	5.78 ^b^	7.34 ^a^	4.66 ^b^	4.85 ^b^	5.07 ^b^	0.40	<0.01	<0.01	<0.01
pg/mg prot.	Jejunum	4.69 ^ab^	4.67 ^ab^	4.25 ^ab^	3.96 ^b^	5.17 ^a^	4.18 ^b^	0.23	<0.01	0.70	0.14
	Ileum	10.42 ^b^	10.44 ^b^	12.97 ^a^	10.34 ^b^	6.68 ^c^	6.26 ^c^	0.23	<0.01	0.04	<0.01
IL-2	Duodenum	4.50	5.32	4.78	5.01	4.69	4.52	0.27	0.45	0.91	0.36
pg/mg prot.	Jejunum	5.10	5.32	4.86	5.34	5.04	3.79	0.51	0.30	0.15	0.36
	Ileum	8.45 ^b^	7.88 ^b^	10.04 ^a^	7.36 ^b^	7.02 ^b^	7.47 ^b^	0.41	0.01	0.09	0.06
IL-4	Duodenum	1.95 ^a^	1.96 ^a^	1.55 ^b^	1.47 ^b^	1.41 ^b^	1.26 ^b^	0.11	<0.01	<0.01	<0.01
pg/mg prot.	Jejunum	1.65	1.73	1.81	1.79	1.83	1.82	0.16	0.27	0.91	0.89
	Ileum	2.63	2.64	2.70	2.33	1.88	2.05	0.23	0.08	0.02	0.02

IL-1β, interleukin 1 beta; IL-2, interleukin 2; IL-4, interleukin 4. ^abcd^ Different letters in the same row denote significant differences between treatments.

**Table 8 biology-12-01390-t008:** Effects of AOP on the expression of immune-related genes in the spleen of broilers.

Items	AOP Supplemental Level (mg/kg)					CTC (mg/kg)		*p*-Value		
	0	250	500	750	1000	50	SEM	ANOVA	Linear	Quadratic
21 d										
*TLR4*	1.00 ^b^	1.13 ^ab^	1.07 ^ab^	1.17 ^ab^	1.38 ^a^	1.21 ^ab^	0.09	0.05	0.01	0.02
*MyD88*	1.00 ^c^	1.01 ^c^	1.24 ^bc^	1.46 ^ab^	1.56 ^a^	1.39 ^ab^	0.10	<0.01	<0.01	<0.01
*NF-κB p50*	1.00 ^b^	1.07 ^ab^	1.10 ^ab^	1.11 ^ab^	1.20 ^a^	1.23 ^a^	0.06	0.01	0.03	0.10
*P38 MAPK*	1.00	0.97	0.95	1.08	1.19	1.13	0.06	0.26	0.06	0.06
*JNK*	1.00 ^ab^	0.90 ^b^	1.09 ^ab^	1.27 ^a^	0.60 ^c^	0.51 ^c^	0.07	<0.01	0.93	0.20
*IL-1β*	1.00 ^b^	1.05 ^b^	1.07 ^b^	1.20 ^ab^	1.52 ^a^	1.16 ^ab^	0.11	0.04	<0.01	<0.01
42 d										
*TLR4*	1.00 ^b^	1.55 ^a^	1.35 ^a^	1.26 ^ab^	1.30 ^a^	1.54 ^a^	0.08	<0.01	0.11	0.02
*MyD88*	1.00 ^b^	1.29 ^ab^	1.19 ^ab^	1.12 ^ab^	1.37 ^a^	1.28 ^ab^	0.09	0.05	0.07	0.20
*NF-κB p50*	1.00 ^b^	1.14 ^ab^	1.09 ^ab^	1.26 ^ab^	1.32 ^a^	1.33 ^a^	0.08	0.05	0.01	0.02
*P38 MAPK*	1.00 ^b^	1.55 ^a^	1.29 ^a^	1.37 ^a^	1.27 ^a^	1.33 ^a^	0.08	<0.01	0.08	0.01
*JNK*	1.00 ^c^	1.15 ^bc^	1.45 ^ab^	1.72 ^a^	1.46 ^ab^	1.68 ^a^	0.09	0.01	<0.01	<0.01
*IL-1β*	1.00 ^c^	1.29 ^bc^	1.38 ^abc^	1.72 ^bc^	1.98 ^a^	1.79 ^ab^	0.09	0.02	<0.01	0.01

*TLR4*, toll-like receptor 4; *MyD88*, myeloid differentiation factor 88; *P38 MAPK*, P38 mitogen-activated protein kinase; *JNK*, C-Jun N-terminal kinase; *NF-κB p50*, nuclear factor kappa B p50; *IL-1β*, interleukin 1 beta. ^abc^ Different letters in the same row denote significant differences between treatments.

**Table 9 biology-12-01390-t009:** Effects of AOP on the expression of immune-related genes in the duodenum of broilers.

Items	AOP Supplemental Level (mg/kg)					CTC (mg/kg)		*p*-Value		
	0	250	500	750	1000	50	SEM	ANOVA	Linear	Quadratic
21 d										
*TLR4*	1.00 ^b^	1.62 ^a^	1.73 ^a^	1.61 ^a^	1.44 ^ab^	1.68 ^a^	0.13	0.02	0.04	<0.01
*MyD88*	1.00 ^b^	1.56 ^a^	1.53 ^a^	1.59 ^a^	1.62 ^a^	1.78 ^a^	0.11	0.01	0.01	<0.01
*NF-κB p50*	1.00 ^b^	1.02 ^b^	1.28 ^a^	1.14 ^ab^	1.11 ^ab^	1.29 ^a^	0.06	0.04	0.13	0.09
*P38 MAPK*	1.00 ^c^	1.21 ^bc^	1.12 ^bc^	1.31 ^abc^	1.63 ^a^	1.39 ^ab^	0.09	0.01	<0.01	<0.01
*JNK*	1.00 ^b^	1.09 ^b^	1.11 ^b^	1.33 ^ab^	1.29 ^ab^	1.51 ^a^	0.12	0.02	0.01	0.03
*IL-1β*	1.00 ^b^	1.49 ^ab^	1.44 ^ab^	1.54 ^ab^	1.93 ^a^	1.79 ^a^	0.13	0.01	<0.01	<0.01
42 d										
*TLR4*	1.00 ^b^	1.37 ^ab^	1.67 ^a^	1.43 ^a^	1.61 ^a^	1.52 ^a^	0.11	0.01	0.01	0.01
*MyD88*	1.00 ^c^	1.19 ^bc^	1.44 ^abc^	1.64 ^ab^	1.68 ^a^	1.89 ^a^	0.13	0.01	<0.01	<0.01
*NF-κB p50*	1.00 ^bc^	0.95 ^c^	1.10 ^abc^	1.25 ^abc^	1.31 ^ab^	1.39 ^a^	0.09	0.02	<0.01	<0.01
*P38 MAPK*	1.00	0.82	0.89	0.97	0.83	1.01	0.05	0.05	0.22	0.38
*JNK*	1.00 ^c^	1.03 ^bc^	1.11 ^bc^	1.28 ^b^	1.16 ^bc^	1.53 ^a^	0.08	<0.01	0.03	0.07
*IL-1β*	1.00 ^b^	1.18 ^b^	1.24 ^b^	1.36 ^ab^	1.75 ^a^	1.30 ^ab^	0.12	0.05	<0.01	0.01

*TLR4*, toll-like receptor 4; *MyD88*, myeloid differentiation factor 88; *P38 MAPK*, P38 mitogen-activated protein kinase; *JNK*, C-Jun N-terminal kinase; *NF-κB p50*, nuclear factor kappa B p50; *IL-1β*, interleukin 1 beta. ^abc^ Different letters in the same row denote significant differences between treatments.

**Table 10 biology-12-01390-t010:** Effects of AOP on the expression of immune-related genes in the jejunum of broilers.

Items	AOP Supplemental Level (mg/kg)					CTC (mg/kg)		*p*-Value		
	0	250	500	750	1000	50	SEM	ANOVA	Linear	Quadratic
21 d										
*TLR4*	1.00	0.91	0.83	0.97	1.11	1.00	0.13	0.75	0.48	0.25
*MyD88*	1.00	1.09	1.22	0.94	1.01	0.98	0.13	0.69	0.54	0.48
*NF-κB p50*	1.00	1.09	1.02	0.98	0.90	1.02	0.06	0.47	0.16	0.18
*P38 MAPK*	1.00	0.93	1.01	0.83	0.82	0.85	0.08	0.42	0.09	0.23
*JNK*	1.00	0.98	1.14	0.99	1.06	1.03	0.10	0.88	0.71	0.89
*IL-1β*	1.00 ^bc^	1.38 ^ab^	1.69 ^a^	1.08 ^bc^	0.96 ^bc^	0.62 ^c^	0.14	<0.01	0.83	0.05
42 d										
*TLR4*	1.00 ^c^	1.32 ^bc^	1.63 ^b^	2.30 ^a^	1.38 ^bc^	1.47 ^b^	0.13	<0.01	0.01	<0.01
*MyD88*	1.00 ^d^	1.23 ^cd^	2.09 ^b^	2.67 ^a^	1.62 ^c^	1.45 ^c^	0.12	<0.01	<0.01	<0.01
*NF-κB p50*	1.00 ^b^	1.23 ^ab^	1.09 ^ab^	1.29 ^a^	1.07 ^ab^	1.05 ^b^	0.06	0.05	0.42	0.13
*P38 MAPK*	1.00 ^abc^	1.13 ^ab^	1.00 ^abc^	1.17 ^a^	0.98 ^bc^	0.82 ^c^	0.05	<0.01	0.99	0.54
*JNK*	1.00 ^b^	1.96 ^a^	2.29 ^a^	2.44 ^a^	2.01 ^a^	2.02 ^a^	0.18	<0.01	<0.01	<0.01
*IL-1β*	1.00 ^c^	1.56 ^abc^	1.85 ^ab^	2.08 ^a^	1.68 ^abc^	1.25 ^bc^	0.19	0.03	0.01	<0.01

*TLR4*, toll-like receptor 4; *MyD88*, myeloid differentiation factor 88; *P38 MAPK*, P38 mitogen-activated protein kinase; *JNK*, C-Jun N-terminal kinase; *NF-κB p50*, nuclear factor kappa B p50; *IL-1β*, interleukin 1 beta. ^abcd^ Different letters in the same row denote significant differences between treatments.

**Table 11 biology-12-01390-t011:** Effects of AOP on the expression of immune-related genes in the ileum of broilers.

Items	AOP Supplemental Level (mg/kg)					CTC (mg/kg)		*p*-Value		
	0	250	500	750	1000	50	SEM	ANOVA	Linear	Quadratic
21 d										
*TLR4*	1.00 ^b^	1.01 ^b^	1.35 ^a^	1.11 ^ab^	1.01 ^b^	1.25 ^ab^	0.08	0.04	0.62	0.12
*MyD88*	1.00 ^b^	1.18 ^ab^	1.32 ^a^	1.31 ^a^	1.05 ^ab^	1.07 ^ab^	0.08	0.05	0.53	0.01
*NF-κB p50*	1.00 ^b^	1.26 ^ab^	1.36 ^a^	1.39 ^a^	1.51 ^a^	1.41 ^a^	0.09	0.02	<0.01	<0.01
*P38 MAPK*	1.00	0.96	1.04	1.06	0.92	1.02	0.08	0.78	0.81	0.65
*JNK*	1.00 ^a^	1.01 ^a^	0.95 ^a^	1.04 ^a^	0.45 ^b^	0.51 ^b^	0.06	<0.01	<0.01	<0.01
*IL-1β*	1.00 ^b^	1.29 ^b^	2.01 ^a^	1.60 ^ab^	1.29 ^b^	1.50 ^ab^	0.17	0.05	0.17	0.02
42 d										
*TLR4*	1.00 ^b^	1.05 ^ab^	1.04 ^ab^	1.26 ^a^	1.26 ^a^	1.14 ^ab^	0.05	0.04	0.03	0.01
*MyD88*	1.00 ^c^	1.32 ^bc^	2.05 ^a^	1.58 ^ab^	1.57 ^ab^	1.63 ^ab^	0.14	0.01	0.02	0.01
*NF-κB p50*	1.00 ^c^	1.15 ^bc^	1.42 ^a^	1.23 ^ab^	1.17 ^bc^	1.29 ^ab^	0.06	0.01	0.07	<0.01
*P38 MAPK*	1.00	1.02	1.11	1.22	1.13	1.29	0.10	0.29	0.08	0.18
*JNK*	1.00	1.09	0.92	1.01	0.95	1.12	0.13	0.89	0.63	0.89
*IL-1β*	1.00 ^c^	1.02 ^c^	1.41 ^ab^	1.61 ^a^	1.28 ^bc^	1.17 ^bc^	0.08	<0.01	<0.01	0.01

*TLR4*, toll-like receptor 4; *MyD88*, myeloid differentiation factor 88; *P38 MAPK*, P38 mitogen-activated protein kinase; *JNK*, C-Jun N-terminal kinase; *NF-κB p50*, nuclear factor kappa B p50; *IL-1β*, interleukin 1 beta. ^abc^ Different letters in the same row denote significant differences between treatments.

## Data Availability

Not applicable.

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
