# Peer review of "Dietary Artemisia Ordosica Polysaccharide Enhances Spleen and Intestinal Immune Response of Broiler Chickens"

_biology, 2023, doi:10.3390/biology12111390_

Round 1

Reviewer 1 Report

This paper describes the effects of dietary supplementation of AOP on growth performance, spleen and small intestine immune function in broilers. In general, I found the manuscript adequately referenced, and insightful.  However, there are many repetitive sentences across the manuscript and needs to be largely reworked. 

 Introduction

The main objective should be coupled with the hypothesis of the study and should be clearly stated in the introduction.

 Material and Methods

The methods information should be adequately provided. Selection criteria of birds for sample collection was not mentioned. Birds were euthanized by cervical dislocation immediately without considering deep plan of sedation and anaesthesia first?. One of the disadvantage of cervical dislocation method if not handled carefully, is that can lead into birds struggle especially when birds are heavy, like broiler chickens (produced for meat). Essential information regarding samples collection and preparation is missing. For instance, which part of the spleen, duodenum, jejunum and ileum was taken? Please add information about the location, size of the tissues collected! Please clarify. Additionally, the variation between individual birds should be assessed, or at least taken into consideration.  The method of sample preparation (e.g., equal number/weight of the sample from each bird being pooled) must be provided.

 Discussion

GENERALLY; I can’t follow the discussion section. It needs to be structurally improved. The mechanism of how 750 mg/kg AOP addition in broiler diets could significantly improve the growth performance is not clear! How such level addition posted the birds performance and led to more body weight gain in comparison to the other levels! The authors did not even consider morphological analysis for intestinal morphological examinations to look into villus growth/ height across different intestinal segments for each dietary treatment.  Which of the resident microbiota aids in determining the ability of the host to harvest energy from its diet. Why it was not cosidered to assess the impact of microbial populations on feed efficiency.

Reviewer 2 Report

This study was conducted to investigate the effects of dietary supplementation with AOP on the immune response in the spleen and small intestine of broiler chickens and to provide a theoretical basis for the application of AOP as an antibiotic alternative. The revised version of the article includes responses and corrections to almost all comments from the previous review. The article requires another correction. The list of proposed/required changes is provided below:

General comments:

For significance use lowercase "p" in italic instead of uppercase "P" throughout the main article

In references volumen number must be in italic

Detailed comments:

L19 broiler chickens instead of broiler

L28 Arbor Acres or Arbor Acre?

L34 in relation to 1000 mg/kg AOP also

In table 1 „Metabolizable energy” not Metabolic energy

L142 4 °C, add a space after the „4”

L174, 176 - and 250 mg/kg AOP also

L187 1000 mg/kg AOP also

L191 250 and 500 mg/kg AOP also

L214 and 250 mg/kg AOP

L222 and 750 and 1000 mg/ kg AOP

L238 and 250 and 500 mg/kg AOP also

L240 IL-6 ? missing from the article?

L255 750 mg/kg AOP and CON also

L339 750 and 1000 mg/kg AOP

L341 jejunum - in lowercase

Round 2

Reviewer 1 Report

The authors responded adequately to the raised points . The manuscript has been significantly improved and now permits publication in Biology.

Author Response

Thank you very much for reviewing our manuscript and also thanks for the very helpful suggestions on improving the manuscript.